# Compromised Blood-Brain Barrier Junctions Enhance Melanoma Cell Intercalation and Extravasation

**DOI:** 10.3390/cancers15205071

**Published:** 2023-10-20

**Authors:** Federico Saltarin, Adrian Wegmüller, Leire Bejarano, Ece Su Ildiz, Pascale Zwicky, Andréj Vianin, Florentin Spadin, Klara Soukup, Vladimir Wischnewski, Britta Engelhardt, Urban Deutsch, Ines J. Marques, Martin Frenz, Johanna A. Joyce, Ruth Lyck

**Affiliations:** 1Theodor Kocher Institute, University of Bern, 3012 Bern, Switzerland; federico.saltarin@outlook.com (F.S.); pascale.zwicky@weizmann.ac.il (P.Z.);; 2Department of Oncology, University of Lausanne, 1011 Lausanne, Switzerlandvladimir.wischnewski@web.de (V.W.);; 3Ludwig Institute for Cancer Research, University of Lausanne, 1011 Lausanne, Switzerland; 4Department of Developmental Biology and Regeneration, Institute of Anatomy, University of Bern, 3012 Bern, Switzerland; 5Department for BioMedical Research (DBMR), University of Bern, 3010 Bern, Switzerland; 6Institute of Applied Physics, University of Bern, 3012 Bern, Switzerland; florentin.spadin@haag-streit.com (F.S.);

**Keywords:** melanoma, metastasis, cell extravasation, blood-brain barrier (BBB), endothelial junctions

## Abstract

**Simple Summary:**

The worst outcome of melanoma is the formation of melanoma-brain metastasis, which depends on the successful extravasation of metastatic melanoma cells across the tight blood-brain barrier (BBB). Therefore, a detailed understanding of the role of the BBB barrier properties in melanoma cell extravasation is important for preventing brain metastasis formation. In this study, we use in vitro live cell imaging to show that melanoma cells exclusively use the junctional pathway for intercalation into the BBB. By using a broad-spectrum protease inhibitor in an experiment analysing barrier disruption by melanoma cells, we confirm the role of proteases in the process of intercalation of melanoma cells into the BBB in vitro. These observations underscore the role of the BBB junctions in the process of melanoma-brain metastasis formation. Finally, using two different in vitro model systems and one in vivo mouse model, we showed that compromised BBB barrier properties facilitate melanoma cell extravasation. Taken together, our data suggest that preserving BBB integrity is an important measure to limit the formation of melanoma-brain metastasis.

**Abstract:**

Melanoma frequently metastasises to the brain, and a detailed understanding of the molecular and cellular mechanisms underlying melanoma cell extravasation across the blood-brain barrier (BBB) is important for preventing brain metastasis formation. Making use of primary mouse brain microvascular endothelial cells (pMBMECs) as an in vitro BBB model, we imaged the interaction of melanoma cells into pMBMEC monolayers. We observed exclusive junctional intercalation of melanoma cells and confirmed that melanoma-induced pMBMEC barrier disruption can be rescued by protease inhibition. Interleukin (IL)-1β stimulated pMBMECs or PECAM-1-knockout (-ko) pMBMECs were employed to model compromised BBB barrier properties in vitro and to determine increased melanoma cell intercalation compared to pMBMECs with intact junctions. The newly generated brain-homing melanoma cell line YUMM1.1-BrM4 was used to reveal increased in vivo extravasation of melanoma cells across the BBB of barrier-compromised PECAM-1-deficient mice compared to controls. Taken together, our data indicate that preserving BBB integrity is an important measure to limit the formation of melanoma-brain metastasis.

## 1. Introduction

Melanoma arises from pigment-producing melanocytes and represents the most lethal form of skin cancer [1]. Patients with advanced melanoma often develop distant metastases to the liver, lung and brain. In particular, the development of brain metastasis is associated with poor prognosis [2,3]. Despite the recent progress in immunotherapy and targeted therapies against melanoma in a subset of patients [2,4,5], there is no broadly effective treatment to tackle established melanoma-brain metastasis.

The formation of metastases is a multi-step process. It starts with local primary tumour invasion and intravasation into the blood or lymphatic vessels. Metastatic cancer cells that manage to survive in the hostile environment of the circulation and to arrest at the capillary level in distant sites gain the opportunity for extravasation and thus access to secondary organs. Once in the organ parenchyma, micrometastases and, eventually, macrometastases can form [6,7]. In the brain, the occurrence of melanoma metastases depends on the successful extravasation across the blood-brain barrier (BBB) [8,9].

The inner layer of the vascular endothelium is formed by a continuous layer of endothelial cells with barrier properties adapted to the function of the organ [10]. The BBB is a particularly tight endothelium with low paracellular permeability that limits the extravasation of cellular components and even the diffusion of ions from the blood into the parenchyma of the central nervous system (CNS) [11]. Adherens and tight junctions represent key molecular structures of the endothelial cell–cell contacts. VE-Cadherin is the major component of the endothelial adherens junctions, while junctional adhesion molecules (JAMs), claudins and TJ-associated MARVEL (MAL and related proteins for vesicle trafficking and membrane link) proteins (TAMPs) [12] together form the endothelial tight junctions [13,14]. Moreover, endothelial junctions include other proteins such as CD99 and platelet endothelial cell adhesion molecule 1 (PECAM-1) [15]. The extraordinary barrier properties of the BBB result in part from the high expression and complex arrangement of claudins and TAMPs [14,16] and thus represent a particular hurdle to the extravasation of cells from the circulation. Therefore, the BBB-damaging effect of therapeutic approaches, such as radiation, should be taken into account with regard to the possible development of new metastases.

Extravasation of melanoma cells is promoted by adhesive interaction between the integrin very late antigen (VLA)-4 on melanoma cells and the immunoglobulin (Ig)-like cell adhesion molecule VCAM-1 on endothelial cells, as previously shown for the BBB [17] and other types of endothelial layers [18,19]. The subsequent diapedesis through the endothelial layer is initiated by the intercalation of melanoma cells into the endothelial cell layer of the BBB and can occur via the paracellular pathway, i.e., between adjacent endothelial cells, or the transcellular pathway, i.e., through a pore across the cytoplasm of an endothelial cell as shown for immune cells [20,21,22] and metastatic cancer cells [23]. Current knowledge of molecular players involved in melanoma cell extravasation has recently been reviewed [24]. In vitro studies revealed that the process of melanoma cell diapedesis is slower compared to immune cell diapedesis and that intercalation into the endothelial monolayer temporally precedes full diapedesis [17]. The limited information available regarding the pathway of melanoma cell diapedesis across the BBB indicates a preference for the paracellular mechanism [23,25,26].

The barrier properties of BBB endothelial cell junctions can be affected by inflammatory stimuli or changes in their molecular composition [22,27,28]. In previous research, we have shown that barrier properties of a tight in vitro BBB model formed by primary mouse brain microvascular endothelial cells (pMBMECs) are compromised upon IL-1β stimulation, compared to TNF-α stimulation [22] or by the lack of PECAM-1 [29].

The current study is based on the hypothesis that the barrier properties of the BBB play an important role in the extravasation process of melanoma cells and, thus, in the formation of melanoma-brain metastases. Therefore, we investigated the impact of compromised endothelial junctions on the intercalation of melanoma cells into the tight BBB. We employed the established pMBMECs as an in vitro model of the BBB [30,31]. For melanoma cells, we utilised the cell lines B78chOVA [17,32], YUMM1.1 [33] and YUMM1.1-BrM4, which is a new brain metastasis-forming derivative of the YUMM1.1 melanoma cells. We used in vitro live cell imaging over time to study the pathway of intercalation into the pMBMECs. Then, we examined the effect of inflammation- or PECAM-1-ko-compromised pMBMECs on melanoma cell adhesion and intercalation. Using trans-endothelial electrical resistance (TEER) measurements, we tested whether protease inhibition can protect the barrier properties of pMBMECs from melanoma-induced disruption. Finally, we challenged our in vitro results by an in vivo analysis of intra- versus extravascular melanoma cell localisation in the brains of PECAM-1-ko or PECAM-1-wild type (wt) mice. The results of this project shed light on the importance of BBB integrity in limiting the formation of melanoma brain metastases.

## 2. Materials and Methods

A flowchart illustrating the methodology is shown in Appendix A.

### 2.1. Mice

All animals used in this study were in the C57BL/6J background. PECAM-1-ko mice were backcrossed for more than ten generations from the previously described PECAM-1-ko mice (PECAM1^tm1Mak^) [34]. PECAM-1-ko mice were crossed with LifeAct-GFP^+^ mice [35] to generate homozygous PECAM-1-ko mice carrying one LifeAct-GFP transgene. VE-CadGFP^+^ (Cdh5^tm9Dvst^) mice were described before [29]. Mouse procedures and experiments were performed in accordance with the Swiss legislation on the protection of animals under the permit number BE77/18, issued by the veterinary office of Canton Bern, Switzerland, for mouse keeping and termination required for the isolation of pMBMECs, and the permit number VD3314, issued by the veterinary office of the Canton Lausanne, Switzerland, for mouse housing, the in vivo intra-cardiac injection of melanoma cells and animal sacrifice for analysis of the in situ localisation of melanoma cells in mouse brains.

### 2.2. BBB Endothelial Cells

Primary mouse brain microvascular endothelial cells (pMBMECs). Isolation of pMBMECs from brains of 6 to 10-week-old mice and culture of pMBMECs were performed as described before [30,31,36]. pMBMECs were seeded directly into the culture vessel required for the respective experiments and thus used without passage.

Brain-derived endothelioma cell lines bEnd5 and bEnd.PECAM-1.2 were described previously [28,31].

### 2.3. Melanoma Cells

B78chOVA melanoma cells. The B78chOVA mouse melanoma cells were a kind gift of Professor Matthew Krummel (University of California, San Francisco, CA, USA) and described previously [17,32].

YUMM1.1 melanoma cells. The YUMM1.1 mouse melanoma cell line was kindly provided by Prof. Ping-Chih Ho (University of Lausanne, Lausanne, Switzerland) and described in detail previously [33].

YUMM1.1-BrM4 melanoma cells. The YUMM1.1-BrM4 melanoma cell line was established by subjecting the parental YUMM1.1 cells to four sequential rounds of in vivo passaging. In each round, melanoma cells were intracardially injected into C57BL/6 mice. When metastatic foci were detected by magnetic resonance imaging, the tumour-bearing brains were digested, and melanoma cells were expanded in vitro for the next round of in vivo selection.

Culture of melanoma cells. B78chOVA, YUMM1.1 and YUMM1.1-BrM4 were cultured in DMEM/F-12 (1 g/L D-Glucose-Glutamine, pyruvate, phenol red, Thermo Fisher Scientific, Rheinach, Switzerland), 10% FBS (Thermo Fisher Scientific, Rheinach, Switzerland), 1% P/S, 1% non-essential amino acids (Thermo Fisher Scientific, Rheinach, Switzerland), in a humidified cell culture incubator at 5% CO_2_. YUMM1.1 and YUMM1.1-BrM4 lines were passaged at ratios between 1 to 3 and 1 to 5; B78chOVA were passaged at a 1 to 10 ratio. All experiments were performed with melanoma cells within ten rounds of passaging. Melanoma cells were tested for lack of mycoplasma contamination using the ScienCell™ Mycoplasma PCR Detection Kit (Chemie Brunschwig, Basel, Switzerland).

### 2.4. Flow Cytometry

Cell surface expression of integrins on melanoma cells was determined by flow cytometry, as described before [17]. Washing steps were performed in PBS. Cells were not fixed before or after staining, which ensures that antigens are only stained if outside of the intact cell membrane. The fluorescence signals were measured using an Attune NxT flow cytometer (Thermo Fisher Scientific, Rheinach, Switzerland).

### 2.5. Antibodies and Cytokines

Primary antibodies. The monoclonal rat anti-mouse VCAM-1 antibody 9DB3 was described previously [22]. The polyclonal goat anti-mouse PECAM-1 antibody AF3628 from R&D, the monoclonal mouse anti-mouse β-actin antibody A5316 from Sigma-Aldrich and the polyclonal rabbit anti-GFP antibody A-11122 from Thermo Fisher Scientific were used for western blotting. The following directly labelled monoclonal antibodies were used for flow cytometry: eFluor 450-M17/4 to CD11a (integrin αL) (eBioscience™, Thermo Fisher Scientific, Rheinach, Switzerland); FITC-M18/4 to CD18 (integrin β2) (eBioscience™, Thermo Fisher Scientific, Rheinach, Switzerland); PerCP-eFluor 710-DATK-32 to integrin α4β7 (LPAM-1) (eBioscience™, Thermo Fisher Scientific, Rheinach, Switzerland); PE-RMV-7 to CD51 (integrin αV) (BioLegend, San Diego, CA, USA); AF647-PS/2 to CD49d (integrin α4) (BioRad, Hercules, CA, USA); APC-eFluor 780-HMb1-1 to CD29 (integrin β1) (eBioscience™, Thermo Fisher Scientific, Rheinach, Switzerland).

Secondary antibodies. For anti-VCAM-1 IF, the Cy3-conjugated AffiniPure Donkey anti-rat IgG (Jackson ImmunoResearch, LuBioScience, Zuerich, Switzerland) was used as the secondary antibody. For western blotting, we made use of the following secondary antibodies: For anti-PECAM-1 WB, we used the Donkey anti-Goat IgG (H + L) Cross-Adsorbed Alexa Fluor^TM^ 680 antibody A-21084 from Thermo Fisher Scientific, for anti-β-actin WB the Donkey anti-Mouse IgG (H + L) Cross-Adsorbed DyLight^TM^ 800 antibody SA5-10172 from Thermo Fisher Scientific and for anti-GFP WB the Donkey anti-Rabbit IgG (H + L) Cross-Adsorbed DyLight^TM^ 800 SA5-10044 from Thermo Fisher Scientific.

Cytokines. Recombinant murine TNF-α was from PromoKine (Vitaris AG, Baar, Switzerland). TNF-α was applied to the primary mouse lung endothelial cells (pMLuECs) or the pMBMECs at 10 ng/mL. Recombinant murine IL-1β was from PeproTech (LubioScience GmbH, Zürich, Switzerland) and was used at 20 ng/mL to stimulate the pMBMECs. Stimulation with TNF-α or IL-1β was for 16 to 20 h before the experiments. Prior to the experiment, pMLuECs or pMBMECs were washed twice (HBSS, 10 mM HEPES, 5% CS) to remove the cytokines.

### 2.6. Western Blot

Protein extracts from confluent endothelial cell cultures were harvested by scraping in ice-cold RIPA buffer (150 mM NaCl, 1% Triton X-100, 0.5% sodium deoxycholate, 0.1% sodium dodecyl sulfate, 50 mM Tris HCl, pH 7.4 plus one tablet per 10 mL of Roche cOmplete™ Mini Protease Inhibitor Cocktail). Protein quantity was determined using the Pierce^TM^ BCA Protein Assay Kit according to the manufacturer’s instructions. Ten µg of each sample were mixed with 5X SDS sample buffer, heated to 95 °C for 5 min and separated by SDS-PAGE (8% resolving gel; 80 V for 20 min and 120 V for 1.5 h). Proteins were transferred to a nitrocellulose membrane (semi-dry blotting at 50 mA for 45 min), which was then blocked for 1 h in Rockland Blocking Buffer MB-070 from Rockland Immunochemicals Inc. (LubioScience GmbH, Zürich, Switzerland). Antibody incubation was done for 1 h at RT or 4 °C overnight. Signals were visualised using an Odyssey Infrared reader (Li-Cor Biosciences, Bad Homburg, Germany) and quantified via Image Studio Lite v4.0 (Li-Cor Biosciences, Bad Homburg, Germany). Background subtraction was done automatically. Protein size was referenced to a prestained protein ladder (Thermo Fisher Scientific, 26619). For calculation of the relative PECAM-1 signal intensity, PECAM-1 signals were normalised to the GFP signal from the LifeAct-GFP protein expressed by the pMBMECs. 

### 2.7. Quantification of VCAM-1 Expression Level

VCAM-1 expression level was determined on pMBMECs with immunofluorescence, which was performed as described before [22]. Briefly, pMBMECs were grown to confluence on µ-slides (ibidi, Vitaris, Baar, Switzerland) and stimulated as indicated. pMBMECs were washed with PBS, fixed with 1% PFA, washed again with PBS, blocked with 5% skimmed milk in PBS complemented with 0.1% Triton-X100 for permeabilisation and then the monoclonal rat anti-mouse VCAM-1 antibody 9DB3 (10 µg/mL) diluted in blocking buffer was added. After an incubation time of 1 to 2 h, the pMBMECs were washed with PBS, and 10 µg/mL Cy3 labelled secondary goat-anti rat antibody (Jackson ImmunoResearch) in blocking solution supplemented with DAPI (1 µg/mL) were added for 30 min. Finally, pMBMECs were washed again and mounted with Mowiol. Either LifeAct-GFP^+^ signal of the pMBMECs or the pMBMECs co-stained with fluorescently labelled phalloidin ensured the confluence of the endothelial layer. In addition, the density of cell nuclei was inspected. Only endothelial layers with comparable density were used for VCAM-1 quantification. Images were acquired for all samples with identical settings with an AxioObserver.Z1 inverted microscope using a Plan-Neofluar 10×/0.3 objective, a camera Axiocam 712 mono and the ZEN 3.4 blue software (Zeiss, Feldbach, Switzerland). VCAM-1 mean signal intensity was determined using the Image J software version 1.53 (NIH, Bethesda, MD, USA). 

### 2.8. Melanoma Cell Interactions with Endothelial Cells In Vitro

Melanoma cell preparation prior to the experiments. For imaging experiments, YUMM1.1 and YUMM1.1-BrM4 melanoma cells were stained with 5 μM CellTracker™ Orange in culture medium for 20 min. B78chOVA were not stained because they are fluorescent by their mCherry expression. Melanoma cells were 1× washed, followed by a 5-min incubation in wash buffer 2 (WB2) (HBSS, 25 mM HEPES, 5 mM EDTA) for detachment. Floating melanoma cells were collected by centrifugation at 200× *g* for 5 min and resuspended at 2 × 10^6^ cells/mL in PBS, 0.1% BSA. Immediately before starting the experiment, the same volume of MAM was added to the melanoma cells, giving rise to 1 × 10^6^ cells/mL.

Melanoma cell adhesion under static conditions. pMBMECs were seeded on Matrigel-coated chambers of a 12-well Ibidi µ-clear slide and grown to confluence. Prior to the experiment, endothelial cells were washed once with pre-warmed MAM, followed by the addition of 100 μL of MAM per well. 100 μL of melanoma cell suspension was added per well and allowed to settle and adhere to the endothelial cells for 10 min. The 12-well chamber was removed, and unbound cells were washed away with PBS. Cells were fixed with 2.5% Glutaraldehyde in PBS for 2 h on ice. Slides were washed twice in PBS and mounted with Mowiol (homemade). Images were acquired for all samples with identical settings using an AxioObserver.Z1 microscope with a Plan-Neofluar 10×/0.3 objective, an Axiocam 712 mono camera and the ZEN 3.4 blue software (Zeiss, Feldbach, Switzerland). Adhered cells were automatically quantified with a FIJI/ImageJ macro (custom-built).

Melanoma cell shear resistant arrest. Shear-resistant arrest of melanoma cells on the apical side of pMBMECs or pMLuECs under flow was performed as previously described [17]. Briefly, ECs were cultured on µ-dishes (ibidi^®^, Vitaris, Baar, Switzerland) until confluence and cytokine-stimulated if indicated. Melanoma cells were perfused over the ECs through a custom-made flow chamber at a concentration of 1 × 10^6^ cells/mL [37] at 0.1 dyn/cm^2^ for 4 min. Then, flow strength was increased to physiological 1.5 dyn/cm^2^. After a period of 1 to 2 min, all non-adherent melanoma cells were washed away, and the number of shear-resistant arrested melanoma cells was determined. Automated image acquisition was started at the beginning of the melanoma cell accumulation using an AxioObserver.Z1 microscope with a Plan-Neofluar 10×/0.3 objective, an Axiocam 712 mono camera and the ZEN 3.4 blue software (Zeiss, Feldbach, Switzerland). Shear resistantly arrested melanoma cells were automatically quantified with a FIJI/ImageJ macro (custom-built).

Melanoma cell intercalation after shear-resistant arrest. The experiment was continued for an additional 90 min in the absence of flow to determine the percentage of intercalation events from shear-resistant arrested melanoma cells. For better resolution, the objective was changed to Plan-Neofluar 40×/0.6 (Zeiss, Feldbach, Switzerland). Imaging was in tiles, and the imaging interval was changed to 6 to 10 min to keep the FOV. Melanoma cell intercalation became visible by the spreading of the melanoma cell and displacement of the VE-CadGFP or the LifeAct-GFP signal from the pMBMECs.

Determination of melanoma cell surface area. The area covered by the spreading melanoma cell during intercalation was determined using the FIJI/ImageJ software version 1.53 by drawing a contour around the melanoma cell in the individual images of the time series.

Quantification of VE-CadGFP signal under the adherent melanoma cell. The mean fluorescence intensity of the GFP channel was determined for the melanoma contour area of each individual image over the intercalation time of 90 min using the FIJI/ImageJ software.

Melanoma cell intercalation in a multi-well setup. Intercalation of melanoma cells into confluent monolayers of pMBMECs with multiple samples in parallel was performed in a 96-well setup as previously described [17]. Imaging in the GFP (LifeAct-GFP) and brightfield channels was performed with the IN-Cell Analyzer 2000 fully automated microscope (GE-Healthcare, Chicago, IL, USA). Three fields of view (FOV) per well were acquired using a Plan Fluor 20×/0.45 extra-long working distance (ELWD) objective (GE-Healthcare, Chicago, IL, USA), with a 5-minute time resolution for a total of 2 h. Intercalation of melanoma cells into the BBB endothelial layer could be detected and quantified through the displacement of the LifeAct-GFP signal. The number of intercalating B78chOVA melanoma cells was evaluated using FIJI/ImageJ. Intercalation of YUMM1.1 melanoma cells into VE-CadGFP^+^ pMBMECs was imaged with an LSM800 confocal microscope (Carl Zeiss), using a Live Cell Imaging (LCI) Plan-Neofluar 25×/0.8 objective (Zeiss, Feldbach, Switzerland), with tiled imaging.

Time-limited adhesive interaction of pMBMECs and B78chOVA melanoma cells to determine PECAM-1 degradation. LifeAct-GFP^+^ pMBMECs were grown to a confluent monolayer in 24 well cell culture plates (bioswisstec Ltd., Schaffhausen, Switzerland). Mouse melanoma cells were detached using EDTA-containing wash buffer (HBSS, 10 mM HEPES, 5 mM EDTA) and collected in PBS, 0.5% BSA. Following centrifugation (200× *g* 10 min), the pellet was resuspended in 0.5 mL PBS, 0.5% BSA. Next, 1.3 × 10^6^ melanoma cells were added to the pMBMECs monolayer. pMBMECs and melanoma cells were then allowed to adhesively interact for 90 min, followed by extensive PBS washing to remove the melanoma cells. pMBMECs proteins were then extracted using RIPA buffer.

### 2.9. Transendothelial Electrical Resistance (TEER)

pMBMECs were grown to a confluent monolayer on filter inserts (0.4 µm pore size and 1 × 10^8^ pores/cm^2^ pore density, ThinCert^TM^, Greiner Bio-One, Vitaris AG, Baar, Switzerland) without TEER-increasing media supplements such as hydrocortisone or cAMP-stabilizing agents. Impedance TEER measurements (CellZscope^®^, Nanoanalytics, Muenster, Germany) were started three days after seeding according to the manufacturer’s instructions. During all measurements, the instrument was placed in the cell-culture incubator to ensure a physiological temperature of 37 °C. The raw TEER values are listed in Appendix A. Assessment of pMBMEC barrier disruption by B78chOVA melanoma in the presence or absence of the MMP inhibitor GM6001 (Abcam plc, Cambridge, UK) was started six to eight days after seeding when TEER developed. The mouse melanoma cells were detached using EDTA-containing wash buffer (HBSS, 10 mM HEPES, 5 mM EDTA) and then collected in PBS, 0.5% BSA. After centrifugation (200× *g* 10 min), melanoma cell pellets were resuspended in PBS, 0.5% BSA at 2.5 × 10^6^ cells/mL. Where indicated, GM6001 (20 µM) or DMSO (1:1000) were added to the pMBMECs and the melanoma cells 15 min prior to the experiment. Then, 5 × 10^4^ melanoma cells in 20 µL were added into the wells with pMBMECs. TEER measurement was started immediately with one measurement every 15 min. TEER values were then exported and evaluated in GraphPad Prism version 9.

### 2.10. Melanoma Cell Binding to Immobilized Proteins

The binding assay of melanoma cells to immobilised proteins under static conditions was performed as described before [17]. For each experiment, a sample in which we checked the lack of binding of melanoma cells to immobilized BSA served as an internal control [17]. Images were acquired for all samples with identical settings with an AxioObserver.Z1 inverted microscope using a Plan-Neofluar 10×/0.3 objective, an Axiocam 712 mono camera and the ZEN 3.4 blue software (Zeiss, Feldbach, Switzerland). The number of bound melanoma cells was quantified in 3 FOVs per sample using Fiji/ImageJ, and the bound cells per mm^2^ were calculated as the average number of cells divided by the FOV area. 

### 2.11. Detection of Melanoma Cells in the Zebrafish Brain In Situ

Approximately 100 to 200 YUMM1.1 or YUMM1.1-BrM4 melanoma cells labelled with CellTracker™ CM-DiI Dye (Thermo Fisher Scientific, Rheinach, Switzerland) were injected into the yolk sac of *casper*;Tg*(fli1a:EGFP)* zebrafish embryos at 2 dpf. Fish larvae were inspected 2 to 3 times per day for three days using the Nikon SMZ25 stereo microscope with a maximal 10× magnification (Nikon Europe B.V., 8132 Egg, Switzerland). For high-resolution imaging, zebrafish larvae were euthanised at 5 dpf, fixed for 2 h at room temperature in 4% PFA, followed by immunofluorescence. Fixed larvae were washed with PBS, 0.1% Tween, permeabilised at room temperature for 1 h with PBS, 1% Triton X-100 and afterwards blocked for 2 h with PBS, 5% BSA. Primary antibody incubation was done with the chicken anti-GFP antibody (1:300, AVES 10-10) overnight at 4 °C. The following day, the embryos were washed with PBS, 0.1% Tween, and incubated with secondary goat anti-chicken IgY (H+L) Alexa Fluor^®^ 488 antibody conjugate (1:250, Life Science, Merck KGaA, Darmstadt, Germany) and DAPI (1:1000, Sigma) for 2 h at room temperature. After this time, larvae were washed with PBS 0.1% Tween and mounted in 1% low melting agarose for imaging. Imaging was done with a Leica digital light sheet TCS SP8 confocal microscope, using an HC PL APO 20×/0.75 IMM CORR CS2 objective in water immersion. 

### 2.12. Detection of Melanoma Cells in the Mouse Brain In Situ

Intracardiac cancer cell injection. YUMM1.1-BrM4 cells were detached and resuspended at 3 × 10^6^ cells/mL in 1 mL of pre-warmed serum-free medium (DMEM-F12). CellTracker™ Deep Red (Thermo Fisher Scientific, Rheinach, Switzerland) was diluted in 1 mL pre-warmed serum-free medium to a 2× working solution (10 µM). Then, the CellTracker™ solution was added to the cells to a final concentration of 5 µM, and the staining solution was incubated at 37 °C for 30 min. Melanoma cells were then washed, filtered with a 100 µm strainer and resuspended to 2.5 × 10^6^ cells/mL. 100 µL of the CellTracker stained melanoma cells were injected intracardially into LifeAct-GFP^+^ mice.

Organ fixation and isolation. Mice were injected with Pentobarbital and perfused with 10 mL PBS, followed by perfusion with 10 mL 2% PFA in PBS. Brains from every mouse were isolated and collected in 2% PFA in PBS for 48 h, followed by washes and storage in PBS.

Confocal Microscopy analysis. The isolated and fixed brains were embedded in 2% low-gelling temperature agarose (Sigma-Aldrich, Merck KGaA, Darmstadt, Germany) to provide tissue stability during the sectioning. 100 µm thick sagittal sections of the brains were cut using a vibrating blade microtome (Leica Microsystems, Wetzlar, Germany) and collected in PBS. Then, the slices were mounted with Mowiol on microscopy slides and stored in the dark overnight. Imaging of the sections was performed with an LSM800 confocal microscope. The Plan-Apo 10×/0.45 objective (Carl Zeiss) was used to obtain full-slices low magnification images, and the Plan-Neofluar 40×/1.3 oil objective was used to acquire high magnification and high-resolution images. The 3D movies were carried out with a Nikon A1 confocal microscope using a 60×/0.95 objective and Nikon NIS-Elements AR 5.30.02 software in 41 z-steps over 40 µm z-distance. The 3D animation was produced with NIS-Elements software using the shaded α blending method.

### 2.13. Image and Data Analysis

Image analysis was performed using Zen Blue software 3.4 (Carl Zeiss), FIJI/ImageJ software and ImageJ macros (National Institute of Health, Bethesda, MD, USA). Data analysis and plotting were performed using Microsoft Excel and GraphPad Prism.

### 2.14. Statistics

If not stated otherwise, error bars show the standard error of the mean and statistical significance was calculated with the unpaired t-test in the GraphPad Prism software version 9 (GraphPad Software, Boston, MA, USA). ns, non-significant. *, *p* value ≤ 0.05, **, *p* value ≤ 0.01, ***, *p* value ≤ 0.005, ****, *p* value ≤ 0.001.

## 3. Results

### 3.1. Cytokine-Induced Junctional Impairment of the BBB Enhances Melanoma Cell Intercalation

To investigate the role of BBB-barrier properties in melanoma cell extravasation, we compared the intercalation of melanoma cells into endothelial layers of an in vitro BBB with different barrier properties induced by two different inflammatory states. As an in vitro BBB model, we used primary mouse brain microvascular endothelial cells (pMBMECs) previously characterised for the formation of BBB-like tight endothelial junctions [30,31]. To stimulate the inflammatory states, we used the cytokines TNF-α and IL-1β. Both cytokines induce upregulation of inflammatory cell adhesion molecules (CAMs), but IL-1β stimulation perturbs the interendothelial cell–cell junctions more than stimulation with TNF-α [22]. To image melanoma cell intercalation, pMBMECs isolated from LifeAct-GFP^+^ transgenic mice (LifeAct-GFP^+^ pMBMECs) [35] and the B78 melanoma cell derivative B78chOVA were used [32]. The spatial displacement of the LifeAct-GFP signal of the pMBMECs combined with the simultaneous spreading of the mCherry expressing melanoma cells, which was visible in phase contrast and the mCherry fluorescence channels, allowed an unambiguous identification of melanoma cell intercalation (Figure 1G and Appendix A) [17]. The melanoma cell intercalation was imaged over a 2-h period in a multi-well-setup, and intercalation events were counted at 30, 60, 90 and 120 min after the start of the experiment. From 60 min after the start of the experiment, significantly more melanoma cells intercalated into IL-1β-stimulated pMBMECs than into TNF-α stimulated pMBMECs (Figure 1A). Thus, more melanoma cells intercalated under the condition with compromised barrier properties of the IL-1β-stimulated pMBMECs.

We further considered the crucial role of vascular cell adhesion molecule (VCAM)-1 in the adhesion of melanoma cells [17,38] and the increased VCAM-1 expression levels on IL-1β-stimulated pMBMECs compared to TNF-α stimulated pMBMECs as observed previously [22], and shown in Figure 1B,C. Therefore, we next analysed whether the enhanced melanoma cell intercalation into IL-1β-stimulated pMBMECs is only the result of enhanced adhesion or is additionally enhanced. To this end, B78chOVA melanoma cells were accumulated on the pMBMECs under low flow conditions, followed by an increased flow pulse to remove non-adherent melanoma cells. The shear-resistant arrest of B78chOVA melanoma cells to IL-1β-stimulated pMBMECs was significantly enhanced compared to TNF-α-stimulated pMBMECs (Figure 1D). To resolve whether the increased melanoma cell intercalation occurred independently of increased melanoma cell adhesion, we followed the intercalation fate of shear-resistant arrested melanoma cells for 90 min (Figure 1E). The use of the flow chamber ensured that only melanoma cells that underwent shear-resistant arrest in the initial period of the experiment were subsequently available for intercalation. For evaluation, the number of arrested melanoma cells was set to 100% under both conditions. We determined that the percentage of melanoma cell intercalation into IL-1β-stimulated pMBMECs was still significantly increased compared to TNF-α-stimulated pMBMECs (Figure 1F).

Taken together, we here demonstrate increased shear-resistant arrest of melanoma cells IL-1β-stimulated pMBMECs compared to the TNF-α stimulated pMBMECs. However, the intercalation of melanoma cells into the junction-compromised Il-1β-stimulated pMBMECs was stronger than the upregulation of VCAM-1 and thus may be a consequence of the weakened junctions of the pMBMECs.

### 3.2. Melanoma Cells Intercalate into pMBMECs Exclusively at Their Cell–Cell Junctions

The observation of increased intercalation of melanoma cells into pMBMECs with compromised junctions prompted us to investigate the localisation of intercalation into pMBMECs in relation to endothelial cell–cell junctions. To address the pathway for BBB intercalation, we made use of VE-cadherin-GFP (VE-CadGFP) expressing pMBMECs as an in vitro model of the BBB with GFP-tagged cell–cell junctions [22]. To analyse the pathway, we employed the flow chamber setup in which only melanoma cells that had shear-resistantly arrested could subsequently intercalate into TNF-α stimulated pMBMECs. The overlay of phase contrast and the GFP signal revealed that the B78chOVA melanoma cell intercalation was consistently associated with the disappearance of VE-CadGFP fluorescence from the endothelial junctions (Figure 2). A temporal analysis of the intercalation process demonstrated the concomitant loss of the junctional endothelial GFP signal and the spread of melanoma cells (Figure 2B,C and Appendix A). To evaluate the pathway of intercalation into unstimulated pMBMECs, we employed a multi-well setup that allowed the inspection of significant numbers of intercalation events. Also, here, all intercalation events were accompanied by loss of the VE-CadGFP signal and, therefore, categorised as junctional.

The exclusive junctional intercalation of the melanoma cells corresponded to our previously observed barrier disruption of pMBMECs by melanoma cells [17]. A protease-mediated mechanism of junction opening during melanoma cell extravasation would be consistent with a previous report on melanoma lung metastasis [39] and was, therefore, our primary hypothesis. This prompted us to perform a pilot experiment, in which we measured the TEER of pMBMECs after the addition of B78chOVA with and without the broad-spectrum matrix metalloproteinase (MMP) inhibitor GM6001 (synonyms: Ilomastat, Galardin) [40,41] over a period of 3 h. A statistical evaluation of the TEER values 3 h after the addition of the melanoma cells confirmed that the TEER formed by pMBMECs was not affected by DMSO or GM6001 alone (Appendix A). As previously described [17], B78chOVA cells caused a large decrease in TEER values at all time points (Appendix A). However, the TEER drop caused by B78chOVA was significantly rescued by the addition of GM6001 (Appendix A).

Taken together, we exclusively observed the junctional pathway of intercalation of B78chOVA into TNF-α or unstimulated pMBMECs and provided evidence for an MMP-driven mechanism of junctional opening. Therefore, we decided to further investigate the role of BBB endothelial cell junctions in the process of melanoma cell extravasation.

### 3.3. YUMM1.1-BrM4 Is a New Brain-Homing Melanoma Cell Line

The use of cell lines that metastasise into the brain parenchyma in vivo is relevant for the study of melanoma cell extravasation across the BBB [42]. Following four rounds of selection for brain metastasis in vivo, we established a variant of the mouse YUMM1.1 melanoma cell line with human-relevant driver mutations [33] that forms brain metastases (here denoted as YUMM1.1-BrM4). Since the integrin VLA-4 is crucial for adhesion to and intercalation into the BBB [17], we performed flow cytometry experiments to assess the immunoreactivity of YUMM1.1-BrM4 cells towards selected integrins. Similar to B78chOVA and the parental YUMM1.1 cell line, YUMM1.1-BrM4 cells showed immunoreactivity to the integrin subunits α4, αV and β1, but no signal for the integrin subunits αL and β2 and the integrin heterodimer α4β7 in flow cytometry analysis (Figure 3A–D). Thus, we concluded that the YUMM1.1-BrM4 can form the VCAM-1 integrin binding partner VLA-4 from the α4 and β1 subunits and one or more αV-integrins, which are ligands to vitronectin, but lack the ICAM-1 integrin binding partner LFA-1 formed by the subunits αL and β2.

We next tested the binding and adhesion of B78chOVA, YUMM1.1, and YUMM1.1-BrM4 to immobilised VCAM-1 or ICAM-1 and to TNF-α or IL-1β stimulated pMBMECs (Figure 3E–J). All three cell lines bound to immobilised VCAM-1 but not to immobilised ICAM-1 (Figure 3E–G). The binding assays also included vitronectin as a positive control and bovine serum albumin (BSA) as a negative control. The binding of all three melanoma cell lines to vitronectin was comparable to VCAM-1 (Figure 3E–G). Similarly, all three melanoma cell lines showed enhanced adhesion to IL-1β-stimulated pMBMECs compared to TNF-α-stimulated pMBMECs, although the data from the YUMM1.1 parental cell line were not significant (Figure 3H–J). Thus, like YUMM1.1 and B78chOVA, YUMM1.1-BrM4 has the molecular machinery to use endothelial VCAM-1 as a ligand and thus to increasingly adhere to pMBMECs with elevated VCAM-1.

To evaluate the overall ability of YUMM1.1-BrM4 melanoma cells to access brain tissue, YUMM1.1-BrM4 melanoma cells were injected into the yolk sac of zebrafish embryos at two days post fertilisation (dpf). The migratory potential of the YUMM1.1-BrM4 melanoma cells was observed daily for three days, until 5 dpf, at which time the larvae were euthanised and imaged (Figure 4). We found that YUMM1.1-BrM4 melanoma cells could be found in the zebrafish head already one day post injection (dpi) (Figure 4A). Using confocal microscopy, we obtained high-resolution images of the brain’s cerebral vessels and determined that melanoma cells were often localised in a perivascular location (Figure 4B).

In summary, the new YUMM1.1-BrM4 cell line forms brain metastases in mice after intracardiac injection, expresses the integrin subunits of VLA-4 and efficiently binds to VCAM-1. Moreover, YUMM1.1-BrM4 cells migrate into the brain of zebrafish larvae after injection into the yolk sac. Thus, we concluded that YUMM1.1-BrM4 represents a physiologically relevant model to study melanoma cell extravasation across the BBB.

### 3.4. Melanoma Cell Intercalation into PECAM-1 Deficient pMBMECs Is Increased

After having observed the exclusive junctional pathway of melanoma cell intercalation, we hypothesised that melanoma cells would also increasingly intercalate into pMBMEC monolayers with compromised junctions but no inflammatory status. As junction-compromised pMBMECs, we employed PECAM-1 deficient (PECAM-1-ko) pMBMECs, which we compared to PECAM-1-wt pMBMECs. PECAM-1-ko pMBMECs maintain junctional localisation of tight junction and adherens junction proteins but show increased permeability to small molecular tracers and establish reduced TEER when compared to PECAM-1-wt pMBMECs [29]. Using immune staining, we determined that the VCAM-1 signal intensity on PECAM-1-ko pMBMECs was comparable to PECAM-1-wt pMBMECs (Figure 5A). Also, YUMM1.1 and YUMM1.1-BrM4 melanoma cells adhered comparably to the apical surfaces of PECAM-1-ko or PECAM-1-wt pMBMECs, respectively (Figure 5B,C). To test whether these observations would also hold true for B78chOVA cells adhesion to a different type of continuous endothelial cells, we used primary mouse lung endothelial cells (pMLuECs). We determined that the monolayer permeability of TNF-α stimulated PECAM-1-ko pMLuECs was increased compared to TNF-α stimulated PECAM-1-wt pMLuECs, while the VCAM-1 expression level was comparable (Appendix A). Adhesion of B78chOVA melanoma cells was tested as shear-resistant arrest, yielding comparable values between PECAM-1-ko and PECAM-1-wt pMLuECs (Appendix A).

Next, we applied in vitro live cell imaging to compare the intercalation of YUMM1.1-BrM4 melanoma cells into PECAM-1-wt or PECAM-1-ko LifeAct-GFP^+^ pMBMECs in multiple samples in parallel. From 30 min after the start of the experiment, we observed significantly increased intercalation of melanoma cells into PECAM-1-ko pMBMECs monolayers compared to PECAM-1-wt pMBMECs (Figure 6A,B) (Appendix A). Also, in the pMLuECs and B78chOVA melanoma cell system, we found increased intercalation of melanoma cells into PECAM-1-ko compared to PECAM-1-wt pMLuECs (Appendix A). We concluded that compromised endothelial cell–cell junctions facilitate the intercalation of melanoma cells into continuous endothelium and particularly into the BBB in vitro.

Given the evident impact of PECAM-1 absence in the pMBMECs on melanoma cell intercalation, we hypothesised that PECAM-1 might be a target of the melanoma cells during the disruption of the pMBMEC monolayer. We evaluated this hypothesis using western blot assays. First, we tested the specificity of the anti-PECAM-1 antibody and observed a clear signal at the expected band size of 130 kDa in protein extracts from brain endothelioma bEnd.5 cells, but an absence of this band from PECAM-1-ko brain endothelioma b.End-PECAM1.2 cells (Figure 7A). Also, B78chOVA melanoma cell protein lysates were negative for any PECAM-1 signal. Next, we assessed the effect of melanoma cells on PECAM-1 of the pMBMECs by adding melanoma cells to LifeAct-GFP^+^ pMBMECs cultures for one hour, followed by whole-cell protein lysis and western blot analysis (Figure 7B). The western blot was probed sequentially for PECAM-1 at 130 kDa and GFP at 30 kDa. Both targets were absent from B78chOVA protein lysates (Figure 7B). The GFP signal intensity of the pMBMECs was then used to normalise for the pMBMEC protein amount without biasing the protein concentration from intercalated melanoma cells. The decrease in PECAM-1 signal intensity for pMBMECs exposed B78chOVA cells was 39% lower than the PECAM-1 signal for pMBMECs without B78chOVA cells (Figure 7C). We concluded that the pMBMECs’ PECAM-1 forms a target for the destruction of the endothelial cell junctions by melanoma cells during the intercalation process.

### 3.5. Melanoma Cell Extravasation across the BBB Is Increased in PECAM-1-ko C57BL/6J Mice In Vivo

Considering the increased intercalation of YUMM1.1-BrM4 melanoma cells into PECAM-1-ko pMBMECs in vitro, we asked whether YUMM1.1-BrM4 melanoma cells also showed increased extravasation across the BBB in PECAM-1-ko C57BL/6J mice in vivo. To this end, fluorescently labelled YUMM1.1-BrM4 melanoma cells were injected intracardially into PECAM-1-wt or PECAM-1-ko LifeAct-GFP^+^ C57BL/6J mice. The LifeAct-GFP^+^ model has previously been shown to delineate the vasculature in the mouse brain [43]. 48 h after melanoma cell injection, mice were sacrificed and perfused to remove intravascular non-adherent melanoma cells and the mouse brains were dissected and fixed with paraformaldehyde (PFA). To address the intra- versus extravascular localisation of melanoma cells in the brain, we imaged the complete volume of three randomly selected 100 µm thick coronal brain slices per mouse (Figure 8A). YUMM1.1-BrM4 cells were identified by their deep red fluorescent stain (Figure 8A–C). The visual inspection of the image stacks allowed us to count, on average, 26 to 27 melanoma cells per brain slice. We assigned melanoma cells to one of two categories: either the melanoma cell was in an intravascular location, which was in the vessel lumen or the lumen with partial intercalation, or the melanoma cell was in an extravascular location, which was outside the green-fluorescent delineation of the brain vasculature. In cases where the localisation of the melanoma cells with respect to the brain vasculature remained unclear, imaging of the respective regions was repeated with higher optical resolution (Figure 8C,E,F) (Appendix A). The combined image analyses showed that the proportion of extravascular melanoma cells was significantly increased in PECAM-1-ko mice compared to PECAM-1-wt mice (Figure 8D). Thus, more melanoma cells crossed the BBB when the endothelial junctions were compromised. These observations thus consolidate our in vitro findings and lead us to the conclusion that compromised BBB junctions allow for increased melanoma cell extravasation to the brain.

## 4. Discussion

A critical event in the progression of melanoma leading to brain metastasis is the access of metastatic cells to the perivascular niche of cerebral vessels [8]. Thus, the successful metastatic melanoma cell must cross the tight BBB. We hypothesised that BBB integrity is a critical parameter in the prevention of melanoma brain metastases and, consequently, that compromised BBB junctions lead to increased extravasation. Therefore, this study focused on investigating the effect of impaired barrier properties of the BBB on the intercalation of melanoma cells into the BBB in vitro and the extravasation of melanoma cells across the BBB in vivo.

To study melanoma cell intercalation into the BBB, we have used a mouse model system of pMBMECs as an in vitro BBB model and syngeneic mouse melanoma cells. Although human BBB endothelial cells and human melanoma cells are available and genetically reflect the human disease, we considered the mouse system advantageous for this project due to the availability of transgenic mouse lines and the ability to also assess melanoma cell extravasation in immunocompetent mice. We have previously studied the BBB characteristics of pMBMECs in detail. pMBMECs closely mirror the RNA transcriptome of naïve brain microvessels, including the expression of BBB solute carriers and efflux transporters [30]. They are responsive to inflammatory stimuli with upregulation of cell adhesion molecules and temporary changes in barrier properties. The adherens junction protein VE-cadherin, the tight junction proteins claudin-5 and occluding, the junctional protein PECAM-1 and the junction-associated proteins ZO-1 and ZO-2 were localised to the cell junctions of pMBMECs [22,29,44]. pMBMECs form moderate but significant TEER without prior stimulation of intracellular cAMP or culture in the presence of hydrocortisone [30,31,36]. pMBMECs have proven valuable in the elucidation of the mechanism of immune cell migration across the BBB [22,29,44] and for studying the role of adhesive cell–cell interactions between melanoma cells and endothelial cells for melanoma cell extravasation across the BBB [17]. Previously, we showed that compromised barrier properties of the pMBMECs can be induced by stimulation with IL-1β [22] or caused by the lack of the junctional protein PECAM-1 [29]. In this study, we showed increased melanoma cell intercalation into pMBMECs afflicted with a barrier-compromised condition compared to the control condition.

In the experiments using cytokine stimulation to induce a compromised barrier condition of the pMBMECs, we had to consider that IL-1β stimulation also upregulates VCAM-1 expression. Endothelial VCAM-1 serves as a ligand for melanoma cell VLA-4 supported adhesion to the endothelial cell surface as shown for human melanoma cell lines MV3 and BLM adhering to immortalised human dermal microvascular endothelial cells (HMEC-1) [18] and for human primary melanoma cells and mouse B78chOVA melanoma cells adhering to pMBMECs [17]. In this previous study, we showed that antibody blocking of VCAM-1 and VLA-4 significantly reduced the intercalation of melanoma cells into the pMBMECs [17]. However, here, we specifically wanted to distinguish between the VCAM-1-promoted intercalation of melanoma cells and a possible additional intercalation-promoting effect of the impaired barrier properties of pMBMECs. By immunofluorescence staining, we here showed a higher VCAM-1 level on IL-1β stimulated pMBMECs compared to the TNF-α stimulated pMBMECs. This finding was in line with previous data achieved with qPCR and On-Cell Western [22]. We further dissected melanoma cell adhesion and intercalation using the flow chamber experiments, which allowed us to closely follow the fate of firmly adhered melanoma cells only. After melanoma cell shear-resistant accumulation, the flow was stopped, and intercalation was recorded. One of our areas of expertise lies in live observation of immune cell extravasation under flow [22,37,45]. However, the observation of melanoma cell extravasation under continuous flow was not successful due to the significantly lower adhesion strength and slower progress of intercalation of melanoma cells compared to effector T cells [22]: Melanoma cell intercalation takes 30 to 90 min (Figure 1 and Figure 2). However, applying blood flow conditions to adherent melanoma cells for this time results in their detachment (unpublished data). Based on this finding and knowing that melanoma cells frequently extravasate at the level of small-diameter capillaries in vivo where blood flow is blocked due to the stuck melanoma cell [8], we considered it appropriate to accumulate the melanoma cells under flow conditions and then observe the dynamics of the interaction in the absence of blood flow conditions. We view a particular strength of this experimental design to be the fact that the observations were limited to melanoma cells that have adhered to the endothelium with adhesive molecular interactions. By setting the number of shear-resistantly arrested melanoma cells to 100% for both conditions, the TNF-α-stimulated and the IL-1β-stimulated pMBMECs, we revealed an adhesion-independent increased melanoma cell intercalation into the barrier-compromised IL-1β-stimulated pMBMECs. These results provided the first evidence for our hypothesis that the barrier properties of the BBB play an important role in the extravasation process of melanoma cells. Thus, inflammation contributes to the extravasation of metastatic melanoma cells through at least two processes: by upregulating trafficking molecules and by weakening endothelial cell–cell junctions. We conclude that the barrier-compromising effect of inflammation must also be considered in the prevention of melanoma cell extravasation.

Using VE-CadGFP^+^ pMBMECs to visualise endothelial cell–cell junctions by in vitro live cell imaging over time, we determined exclusive junctional melanoma cell intercalation into pMBMECs. Intercalation of the melanoma cells occurred in a time frame of 30 to 60 min with concomitant disappearance of the junctional VE-CadGFP signal. The time frame and the junctional route of melanoma cell intercalation occurred in unstimulated, and TNF-α-stimulated pMBMECs. The term "intercalation" for melanoma cells was deliberately chosen to distinguish the observed process from the "diapedesis" event of the T cells. Intercalation is a process by which the melanoma cell appears to squeeze into the endothelial junctions, and the endothelial gap never closes again above the intercalated melanoma cell. Diapedesis is a process in which the T cell crosses the endothelial layer and moves away from the site of diapedesis underneath the endothelial cells. The endothelial gap closes again after T cell diapedesis. We have used VE-CadGFP^+^ pMBMECs before to elucidate the diapedesis pathway of effector T cells [22]. In apparent contrast to melanoma cells, diapedesis of effector T cells across pMBMECs occurred via both pathways, the paracellular and the transcellular route, and was completed within 8 to 10 min [22]. Melanoma cell intercalation into the junctions of human microvascular endothelial cells (HMVECs) or primary rat brain endothelial cells has been shown by immunofluorescence and electron microscopy (EM), respectively [23,25]. A detailed EM analysis presented a precise morphological description of the melanoma cell/endothelial cell interplay, which differed from the endothelial cell/carcinoma cell interplay [23]. While the intercalation process of the melanoma cells disrupts the endothelial layer, the carcinoma cell intercalation results in an intact hybrid endothelial cell/carcinoma cell monolayer [23]. Thus, our observation on the persistent junctional intercalation of melanoma cells into the pMBMEC layer agrees perfectly with the observations made by others. Furthermore, by using live cell imaging of live-stained VE-cadherin-GFP junctions of the pMBMECs and restriction of melanoma cells adhesion to flow-resistant events, we advance knowledge by temporally resolving melanoma cell extravasation and confining extravasation events to those melanoma cells capable of adhering to the pMBMECs under blood flow conditions. Another important implication of the exclusive junctional pathway of intercalation of melanoma cells is the apparent difference from the pathway of T cell diapedesis, which can proceed via both pathways. This gains importance when one considers that VLA-4 is also an important cell adhesion molecule for effector T cells. Therefore, therapeutic exploitation of VLA-4’s role in melanoma cell extravasation would be limited. The knowledge that T cells can also efficiently diapedesis across the BBB endothelial cells along the transcellular route raises the possibility of therapeutically decoupling the inhibition of melanoma cell extravasation from T cell extravasation by BBB tightening. Therefore, maintaining or even strengthening the integrity of the endothelial junctions at the BBB could reduce the formation of melanoma brain metastasis without affecting T cell surveillance of brain tissue.

Using TEER, we demonstrated that melanoma cells disrupt the barrier properties of pMBMECs. This result is consistent with previous observations published by Fazakas and colleagues [25], who reported reduced TEER of rat brain endothelial cells (RBECs) by melanoma cell incubation. The difference between their study and ours relies on the time course of BBB disruption, which may be considered of relatively minor importance. In our previous study [17] and also in this study, we found that the barrier disruption of pMBMECs is already significant within 1 h after the addition of melanoma cells, which is in agreement with the time course of the VE-cadherin-GFP junction disappearance in pMBMECs observed in our live cell imaging experiment. Using the TEER experiment, we rescued the barrier disruption by inhibition of proteases using the broad-spectrum MMP inhibitor GM6001. GM6001 inhibits the invadopodia-specific proteases MMP-2, MMP-9 and MT1-MMP [40,41]. The involvement of MMPs in the progression of melanoma metastases has long been known [46]. It has been found that MMP-2 and -9 are expressed in human melanoma and that MMP-2 is associated with melanoma progression [47]. Several studies have established an association between melanoma MMP expression and extravasation across the BBB. For example, MMP2, produced by melanoma cells after exposure to astrocyte-secreted IL-23, has been shown to promote diapedesis of human melanoma cells in a human BBB model [48]. A specific role of MMP9, produced by melanoma or carcinoma cells, in the in vivo extravasation process across the BBB in the mouse model has been recently described [49]. The cellular structures known to release MMPs in pathological processes are the invadopodia. A role of invadopodia in the extravasation of carcinoma across lung endothelial cell layers or human melanoma across rat brain endothelial cells has been shown previously by targeting the invadopodia drivers cortactin, seprase, Tks4 or Tks5 [25,50,51]. With regard to the role of MMPs in the extravasation of melanoma cells across the BBB, we extend previous knowledge by applying TEER measurements over time in combination with pMBMECs as the BBB model and the MMP inhibitor GM6001 to rescue BBB disruption. Taken together, a potential benefit of protease inhibitors could be early in the course of the disease, before metastases are established, by inhibiting melanoma cell extravasation.

To model melanoma-to-brain metastasis, we derived the brain-homing YUMM1.1-BrM4 cell line by in vivo selection of the parental YUMM1.1 melanoma cells from brain metastatic foci. The YUMM1.1 parental melanoma cell line carries the genetic alterations of BRAF^V600E^ mutation and deletion of phosphatase and tensin homolog (PTEN) and cyclin-dependent kinase inhibitor 2 (CDKN2A) and thus recapitulates the genetic profile of human metastatic melanoma [33]. In addition to brain metastases, YUMM1.1-BrM4 cells occasionally formed cardiac and pericardial lesions, which grew out at the injection site following intracardiac cancer cell delivery into syngeneic mice. More rarely, lesions formed by YUMM1.1-BrM4 cells were also observed in the abdomen, adrenal gland, spine, skin or lungs. The migration of YUMM1.1-BrM4 cells from the yolk sac to the brain region in the zebrafish further supports the brain-specific adaptations of the YUMM1.1-BrM4 cell line. The use of cell lines that preferentially give rise to metastasis to the brain is of fundamental importance for the investigation of brain metastases formation [42]. A recent study identified a sub-population of brain metastasis-initiating breast cancer cells and thus strengthens the importance of appropriate models [52]. In this context, the YUMM1.1-BrM4 derivate of YUMM1.1 represents a relevant model to study the mechanism of melanoma brain metastasis formation.

We determined that both YUMM1.1 and YUMM1.1-BrM4 express VLA-4 on their surface and bind to recombinant VCAM-1 in vitro. Further, both melanoma cell lines do not express LFA-1 and do not bind to ICAM-1. Previously, we reported that this particular integrin profile is also the case for B16F10 melanoma cells and that VLA-4 plays a critical role in BBB breakdown during melanoma cell intercalation [17]. The VLA-4 positive and LFA-1 negative integrin profile of melanoma cells differs from CD4^+^ effector T cells [22], which do express LFA-1 and use the ICAM-1/LFA-1 axis for crawling on the BBB apical face prior to diapedesis [44]. Since melanoma cells do not express LFA-1 and do not crawl prior to intercalation, we identify here another important difference in the extravasation process between melanoma cells and effector T cells. This difference could also be of relevance to a strategy that inhibits melanoma cell extravasation without affecting effector T cell trafficking.

To further investigate the role of compromised cell junctions of the BBB, we selected PECAM-1-ko pMBMECs as a model with impaired BBB barrier properties in the absence of any inflammatory state. PECAM-1 at the endothelial junctions is known to be involved in the regulation of microvascular barrier properties [28,53]. In a previous study, we examined PECAM-1-ko pMBMECs and observed a significantly decreased TEER and increased permeability to a 3 kDa dextran tracer compared to the PECAM-1-wt pMBMECs [29]. In this study, we first confirmed that cell surface expression of VCAM-1 and melanoma cell adhesion were comparable between PECAM-1-ko and PECAM-1-wt pMBMECs. Therefore, we conclude that the increased intercalation of the YUMM1.1-BrM4 melanoma cells into the PECAM-1-ko pMBMECs compared to the PECAM-1-wt pMBMECs is solely caused by the compromised endothelial cell junctions and is independent of an inflammatory state of the pMBMECs or increased melanoma cell adhesion.

Finally, we also assessed YUMM1.1-BrM4 melanoma cell extravasation in the brains of PECAM-1-ko and PECAM-1-wt LifeAct-GFP^+^ mice. We reported significantly increased melanoma cell extravasation in PECAM-1-ko mice compared to PECAM-1-wt mice. We observed single melanoma cells and small clusters of melanoma cells, which we could not clearly distinguish even with DAPI staining since every nucleus in the sample was then stained, making it virtually impossible to distinguish melanoma cell nuclei from the nuclei of other cells. Regarding the study objectives, we concluded that counting the two categories "extravascular" versus "intravascular or intravascular and partially intercalated" would answer the crucial question. However, we also acknowledge that an assessment of single cells versus small cell cluster extravasation using genetically engineered YUMM1.1-BrM4 with a fluorescent protein in the nucleus (e.g., H2B-RFP) would have been interesting to include. As the CNS is a common metastasis site in melanoma patients [54] and melanoma extravasation is an obligatory step for brain colonisation [8], exploring the role of the BBB in vivo is of high relevance. Using PECAM-1-ko mice for studying melanoma extravasation could raise the question about the role of PECAM-1 itself in this process. Voura and colleagues have determined that in vitro, the extravasation process of human melanoma cells across human lung microvascular endothelial cells is independent of molecular interactions involving PECAM-1 [26]. In vivo, one could argue about the effect of lack of PECAM-1 in platelets and the role of blood clot formation for melanoma cell extravasation. Platelet counts were found to be comparable between PECAM-1-ko and control mice [55]. The role of platelet PECAM-1 in thrombus formation in response to stimulation by a variety of agonists has not been conclusively established, as one report describes a rather minor or negligible role [56], while two other studies reported increased thrombus formation in PECAM-1-ko mice [57,58]. It is also debated whether the formation of blood clots is important for the extravasation of metastatic cancer cells. Ward and Martín observed the promotion of cancer cell extravasation by microclots and the associated inflammatory response [59]. On the contrary, Karreman and colleagues have only recently ruled out a driving role of platelet clot formation in the extravasation of carcinoma or melanoma cells across the BBB [49]. Focusing on endothelial PECAM-1, we acknowledge that our data are currently insufficient to conclude that human metastatic melanoma disease is caused by a PECAM-1 variant. However, additional investigations on this topic using multiple single nucleotide polymorphisms (SNPs), along with a focus on PECAM-1 and other endothelial cell junction molecules, would be interesting additions to investigate in future studies.

In conclusion, our study demonstrates that loss of endothelial junction integrity at the BBB facilitates intercalation and extravasation of melanoma cells independent of adhesion. We conclude that any damage to the barrier properties of the BBB can lead to an increased likelihood of developing melanoma brain metastases. Moreover, we propose that inhibition of melanoma cell extravasation might be decoupled from T cell extravasation by BBB tightening. Therefore, maintaining the BBB barrier should be considered in any treatment of patients with metastatic disease.

## 5. Conclusions

In conclusion, our study demonstrates that loss of endothelial junction integrity at the BBB facilitates intercalation and extravasation of melanoma cells independent of adhesion. We conclude that any damage to the barrier properties of the BBB, caused, for example, by irradiation-induced inflammation, can lead to an increased likelihood of developing melanoma brain metastases. Therefore, maintaining the BBB barrier should be considered in any treatment of patients with metastatic disease.

## Figures and Tables

**Figure 1 cancers-15-05071-f001:**
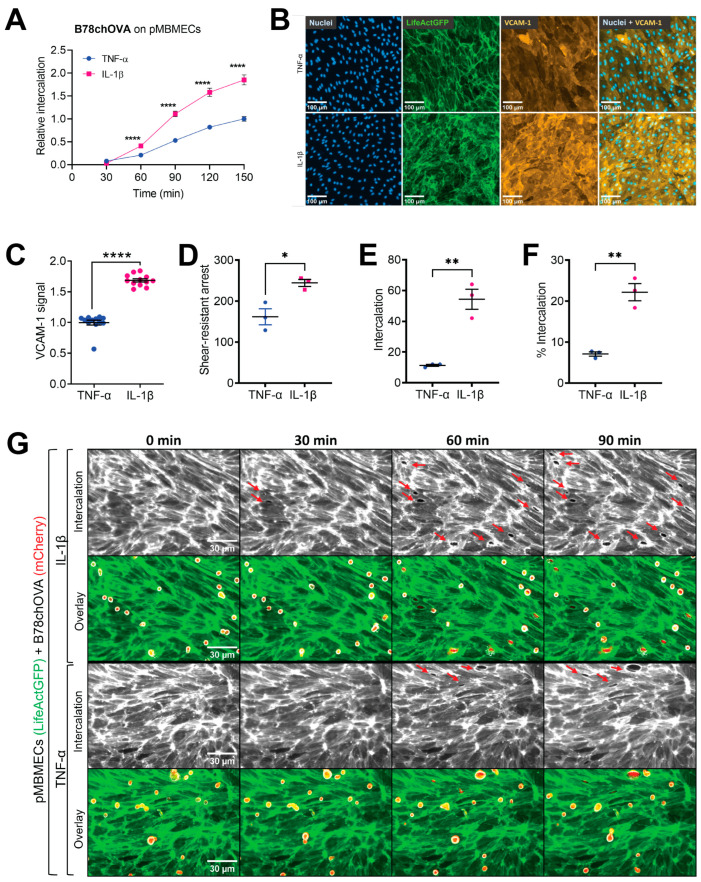
Melanoma cell adhesion to and intercalation into IL-1β or TNF-α stimulated LifeAct-GFP^+^ pMBMECs. (**A**) B78chOVA melanoma cell intercalation into IL-1β stimulated pMBMECs compared to TNF-α stimulated pMBMECs over time expressed as relative values. Data represent the mean of 3 experiments performed in a multi-well format under static conditions in which the first data point was collected 30 min after the addition of the melanoma cells. IL-1β, 11 wells, 48 fields of view (FOVs). TNF-α, ten wells, 42 FOVs. Size of each FOV was 8.8 × 10^4^ µm^2^. P-values refer to the paired time points. For normalisation, intercalation into TNF-α stimulated pMBMECs at 150 min was set to 1.0 in each experiment. Error bars show the standard error of the mean (SEM). (**B**) Immunofluorescence of VCAM-1 on TNF-α or IL-1β stimulated LifeAct-GFP^+^ pMBMECs. Images from left to right, DAPI staining (blue), LifeAct-GFP fluorescence (green), VCAM-1 (orange), and merge from DAPI and VCAM-1 signals. Representative images. (**C**) Relative signal intensity of VCAM-1 immunofluorescence on TNF-α or IL-1β stimulated pMBMECs. Four experiments in triplicates. The mean VCAM-1 fluorescence intensity on TNF-α stimulated pMBMECs was set to 1.0 in each experiment. Error bars, SEM. (**D**) B78chOVA melanoma cell shear-resistant arrest on TNF-α or IL-1β stimulated pMBMECs in absolute numbers per FOV. (**E**) B78chOVA melanoma cell intercalation into IL-1β stimulated compared to TNF-α stimulated pMBMECs 90 min after shear-resistant arrest shown in (**D**). (**F**) B78chOVA melanoma cell intercalation in TNF-α- or IL-1β-stimulated pMBMECs expressed as the per cent of shear-resistant arrested melanoma cells, shown in (**D**). The total number of arrested B78chOVA melanoma cells was set to 100% in both conditions. (**D**–**F**) Size of the FOV, 1.2 × 10^5^ µm^2^. Experiments were performed in triplicates. (**A**,**C**–**F**) *, *p* value ≤ 0.05, **, *p* value ≤ 0.01, ****, *p* value ≤ 0.001. (**G**) Time-lapse imaging of B78chOVA melanoma cell intercalation into TNF-α or IL-1β stimulated pMBMECs after shear-resistant arrest at 0, 30, 60 and 90 min. 1st and 3rd row, displacement of LifeAct-GFP signal of the endothelial cells by melanoma cells over time can be identified as black areas (indicated by the red arrows), which correspond to melanoma cell intercalation sites. The second and fourth rows overlay the LifeAct-GFP signal of the pMBMECs with the phase contrast and mCherry signal to show the positions of B78chOVA melanoma cells. The contrast settings in the phase contrast channel were chosen in order to visualise the higher-contrast melanoma cells. See also Appendix A.

**Figure 2 cancers-15-05071-f002:**
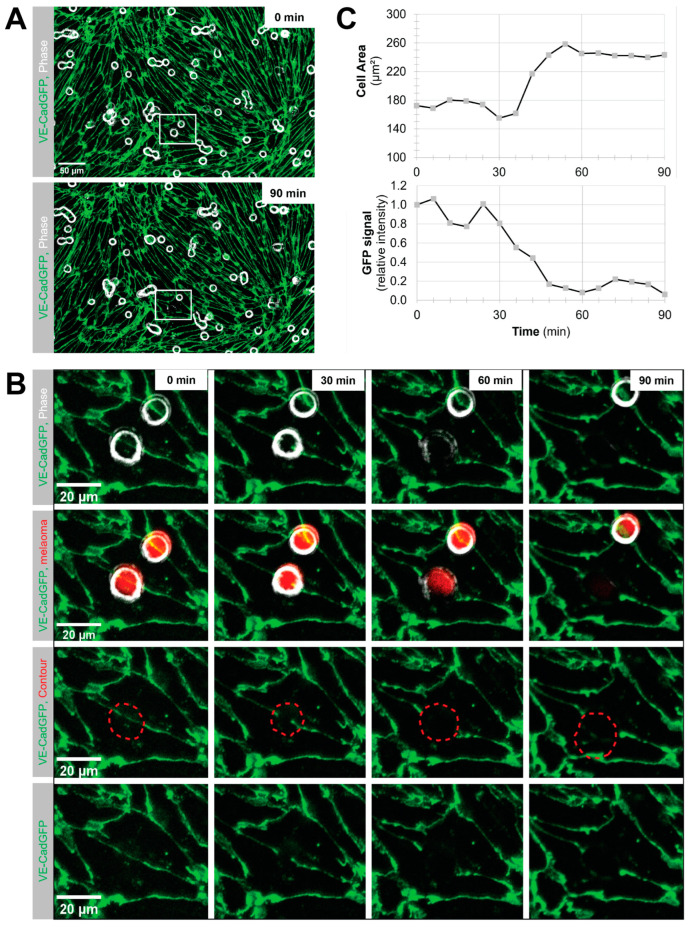
Melanoma cell intercalation at endothelial intercellular junctions. (**A**,**B**). Live cell imaging of B78chOVA melanoma cell intercalation after shear-resistant arrest into TNF-α stimulated VE-CadGFP^+^ pMBMECs over time. Green, VE-CadGFP. Grey, phase contrast adjusted to highlight the melanoma cell spreading. (**A**). Full FOV at 0 and at 90 min time point. Merge of phase contrast and GFP fluorescence. (**B**). Panel with enlarged images corresponding to the area shown with the white rectangle in (**A**) at 0, 30, 60 and 90 min. 0 min corresponds to the moment immediately after the pulse flow that removed non-adherent melanoma cells. Top row, merge of phase contrast and GFP fluorescence. The second row is the same as the top row but with additional red mCherry fluorescence of the melanoma cells. In the third row, GFP fluorescence is overlaid with the red dotted line outlining the contour of the melanoma cell visible in the top row. Bottom row, GFP fluorescence. See also Appendix A. (**C**) Quantification of (top) the area covered by the spreading melanoma cell and (bottom) VE-CadGFP average fluorescence intensity in the red contoured area shown in Figure 1B over time.

**Figure 3 cancers-15-05071-f003:**
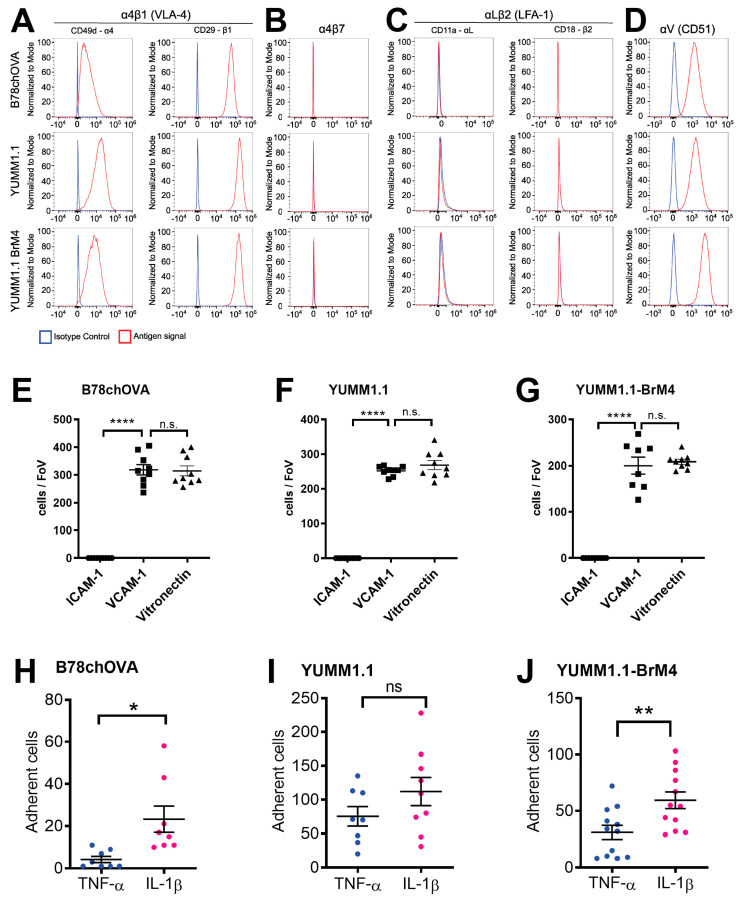
Comparison of YUMM1.1-BrM4 with YUMM1.1 and B78chOVA. (**A**–**D**) Flow cytometry analysis of B78chOVA, YUMM1.1 and YUMM1.1-BrM4 cells for expression of integrins α4 and β1 (**A**), integrin α4β7 (**B**), integrin αL and β2 (**C**) and integrin αV (**D**). Gating was performed to exclude dead and aggregated cells. Shown are representative data from one out of three independent experiments. (**E**–**G**) Binding of B78chOVA (**E**), YUMM1.1 (**F**) and YUMM1.1-BrM4 (**G**) melanoma cells to recombinant mouse ICAM-1, VCAM-1 and Vitronectin under static conditions. Data of the negative control BSA is not shown. Dots correspond to melanoma cells bound per FOV (1.84 × 10^5^ µm^2^). Shown are representative data from one out of 3 independent experiments performed in triplicates with at least 2 FOVs evaluated per sample. (**H**–**J**) Adhesion of B78chOVA (**H**), YUMM1.1 (**I**) or YUMM1.1-BrM4 (**J**) melanoma cells to TNF-α or IL-1β stimulated pMBMECs assessed under static conditions. Values of melanoma cell adhesion to TNF-α stimulated pMBMECs set to 1.0. Three experiments for each type of melanoma cells, at least in duplicate. (**E**–**J**) ns, non-significant. *, *p* value ≤ 0.05, **, *p* value ≤ 0.01, ****, *p* value ≤ 0.001.

**Figure 4 cancers-15-05071-f004:**
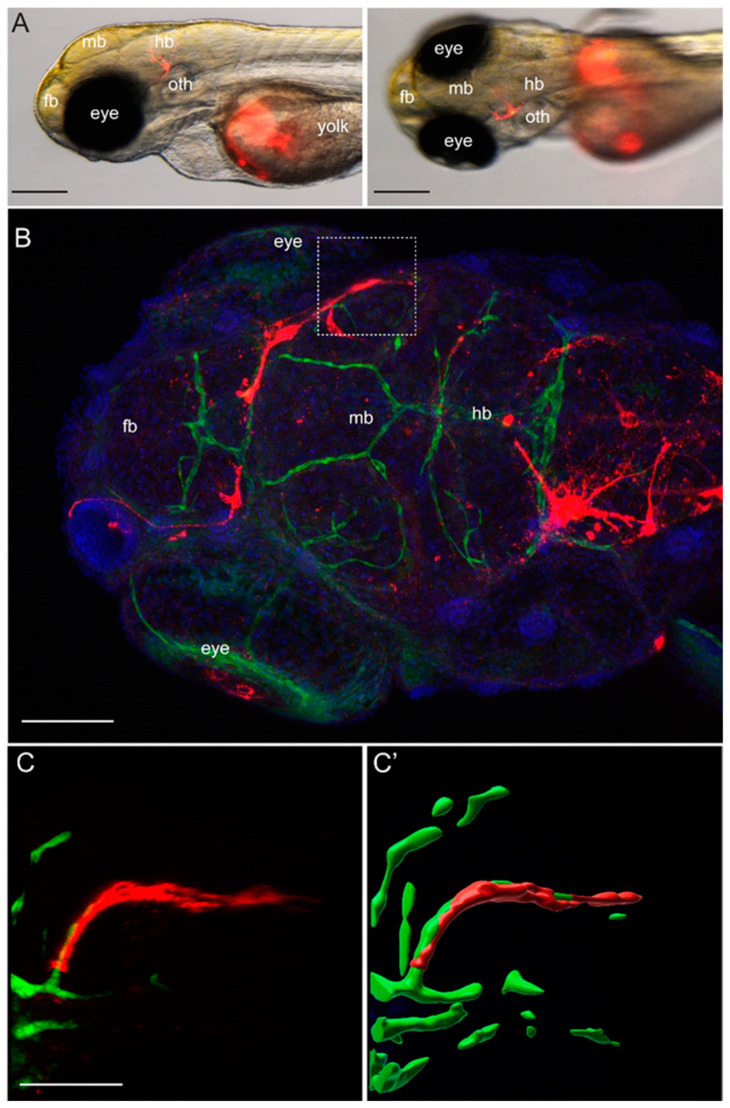
In zebrafish larvae, xenografted YUMM1.1-BrM4 melanoma cells were found outside of blood vessels in the brain region. (**A**) Stereomicroscope images of the head region of a zebrafish larva with a melanoma cell in the brain 2 dpi of the melanoma cells. Left, side view. Right, top view. Scale bar 200 µm. (**B**) Immunofluorescence against GFP in a *casper*;Tg*(fli1a:EGFP)* zebrafish embryo 3 dpi of red-fluorescently labelled YUMM1.1-BrM4 melanoma cells. Top image. 3D reconstruction of the head region in a dorsal view. Green colour marks endothelial cells of the vasculature, red colour labels the YUMM1.1-BrM4 melanoma cells, and blue colour identifies the cell nuclei. Scale bar, 100 µm. (**C**,**C′**) Zoomed view of the boxed area in (**B**). The region shown here has been rotated for better visualisation and shows a lateral view of that area. (**C′**) Surface reconstruction of the image in (**C**). Notice how the YUMM1.1-BrM4 melanoma cells (red) surround the vessel (green). Scale bar 50 µm.

**Figure 5 cancers-15-05071-f005:**
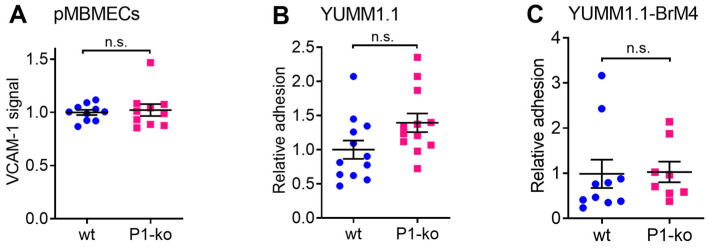
Lack of endothelial PECAM-1 does not affect melanoma cell adhesion on pMBMECs. (**A**) Quantification of immunofluorescence staining of VCAM-1 endothelial cell surface levels on PECAM-1-wt and PECAM-1-ko pMBMECs. (**B**) YUMM1.1 and (**C**) YUMM1.1-BrM4 melanoma cell adhesion on PECAM-1-wt and PECAM-1-ko pMBMECs. (**A**–**C**) PECAM-1-wt set to 1.0 (*n* = 3). n.s., non-significant.

**Figure 6 cancers-15-05071-f006:**
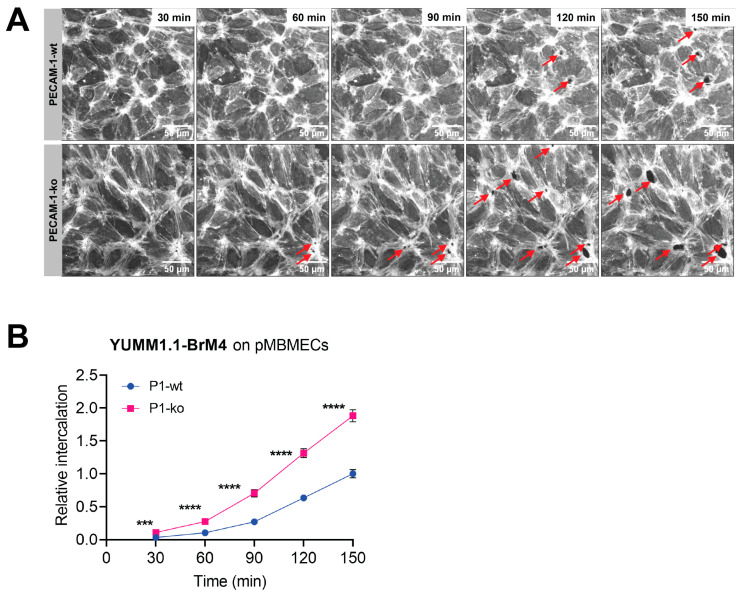
Compromised endothelial junctions enable melanoma cells to intercalate more effectively. (**A**). YUMM1.1-BrM4 melanoma cell intercalation in PECAM-1-wt and PECAM-1-ko LifeAct-GFP^+^ pMBMECs at 30, 60, 90, 120 and 150 min after addition of melanoma cells. Displacement of the LifeAct-GFP signal of the endothelial cells by melanoma cells can be identified as black areas (indicated by the red arrows), which correspond to melanoma cell intercalation sites. Images correspond to data shown in (**B**). (**B**) Quantification of YUMM1.1-BrM4 cell intercalation events over time in PECAM-1-wt and PECAM-1-ko LifeAct-GFP^+^ pMBMECs. Data represent the mean of three experiments performed in triplicates. Intercalation events were counted in 3 FOVs (5.74 × 10^5^ µm^2^) per well. The mean melanoma cell intercalation into PECAM-1-wt at 150 min was set to 1.0. Statistical analysis was performed using the unpaired t-test for significant differences between each time point. (**A**,**B**). The experiments were performed in a 96-well setup for simultaneous data acquisition. ***, *p* value ≤ 0.005, ****, *p* value ≤ 0.001.

**Figure 7 cancers-15-05071-f007:**
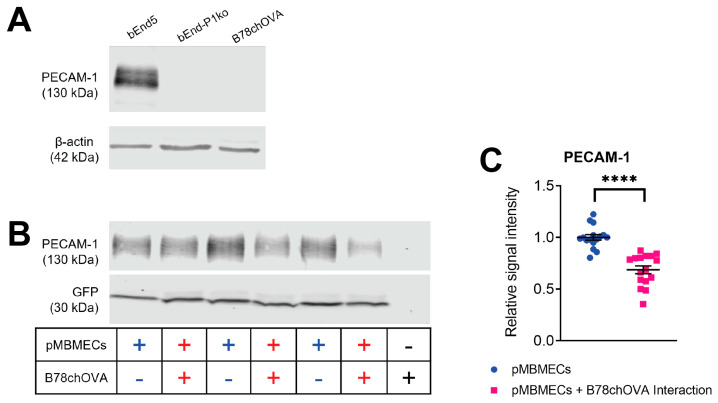
PECAM-1 signal intensity of pMBMECs is lost by incubation with melanoma cells. (**A**) PECAM-1 at 130 kDa and β-actin at 42 kDa were detected by Western blotting protein extracts from bEnd.5 or B78chOVA melanoma cells. Ten µg protein extract was loaded in each lane. (**B**) PECAM-1 at 130 kDa and GFP at 30 kDa were detected by Western blotting protein extracts from LifeAct-GFP^+^ pMBMECs either in a single culture or after the addition of B78chOVA melanoma cells for one hour. Three different samples were loaded each. The right-most lane shows B78chOVA alone. Ten µg protein extracts were loaded in each lane. (**C**) Quantification of the PECAM-1 signal (*n* = 17 from five individual experiments). In each sample, the PECAM-1 protein amount was normalised to the GFP signal. The graph shows each PECAM-1 signal relative to the mean PECAM-1 signal of pMBMECs set to 1.0 in the individual experiments. ****, *p* value ≤ 0.001. Original Western Blots are shown in Appendix A.

**Figure 8 cancers-15-05071-f008:**
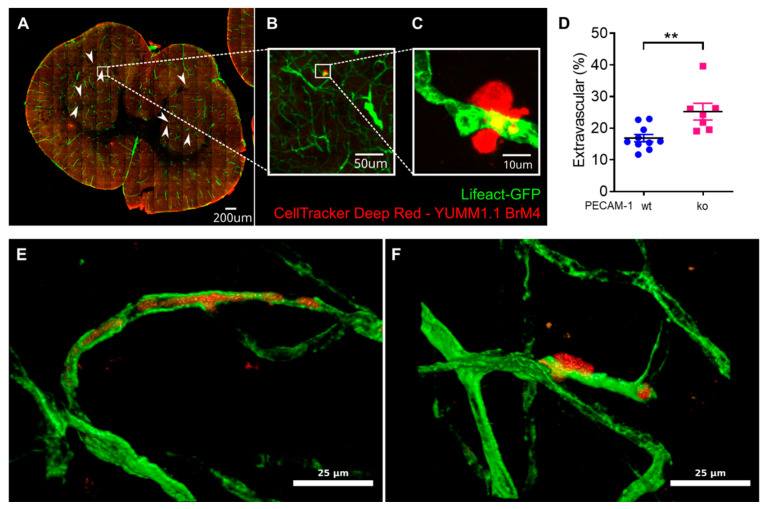
YUMM1.1-BrM4 melanoma cell extravasation across the BBB is increased in vivo in PECAM-1-ko LifeAct-GFP^+^ mice. Analysis of melanoma cell extra- versus intravascular localisation in 100 µm thick brain coronal slices 48 h after melanoma cell intracardial injection in PECAM-1-wt or PECAM-1-ko LifeAct-GFP^+^ C57BL/6J mice in vivo. (**A**) Tiled imaging of a whole brain slice with a 10×/0.45 objective (z-projection from 9 slices). White arrowheads point to melanoma cells. (**B**) Zoom into the image shown in (**A**) showing one extravascular melanoma cell. (**C**) High-resolution imaging (40×/1.4 oil immersion objective) of the extravascular melanoma cell shown in (**B**) (z-projection from 39 slices over 18.3 µm in z). (**D**) Percentages of extravascular cells in PECAM-1-wt (10 mice, three brain slices each, total 805 melanoma cells) and PECAM-1-ko (seven mice, three brain slices each, total 553 melanoma cells) LifeAct-GFP^+^ C57BL/6J mice. **, *p* value ≤ 0.01. (**E**,**F**) Side-by-side comparison of an intravascular (**E**) and an extravascular (**F**) melanoma cell. See also Appendix A.

## Data Availability

Original data will be made available upon request.

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
