# Peer review of "Compromised Blood-Brain Barrier Junctions Enhance Melanoma Cell Intercalation and Extravasation"

_cancers, 2023, doi:10.3390/cancers15205071_

Round 1
Reviewer 1 Report
In the paper entitled “Compromised blood-brain barrier junctions enhance melanoma cell intercalation and extravasation” Saltarin et al. investigate the involvement of tight and adherens junctions of brain endothelial cells in brain metastasis formation.
Unfortunately, the study contains very few novelties in comparison to the literature, it is almost exclusively repetition of already published data, as follows:
- Disruption of the tight junctions of brain endothelial cells and exclusive junctional extravasation of melanoma cells has already been shown in several papers (ref. 22 of the present manuscript is one of the examples). In this respect, the statement that “the role of the BBB barrier properties in melanoma cell extravasation is poorly understood” is an overstatement. In addition, lines 436-439 falsely suggest that this is the first time when this hypothesis emerges.
- Similarly, there is ample in vitro and in vivo evidence of the role of proteases in the melanoma-induced disruption of brain endothelial tight junctions (e.g. a recent review in Cancers: DOI: 10.3390/cancers15082258).
Therefore, no new mechanism is described, the minimal novelty consists in the following:
- It is shown that enhanced melanoma cell intercalation in IL-1β-stimulated PBMECs occurs partly independently of increased VCAM-expression and melanoma adhesion. What is the relevance of this finding?
- Although not a novel mechanism, in vivo evaluation of extravasated melanoma cells in PECAM-1-KO animals is an original element. Question: How were individual melanoma cells differentiated from small cell clusters in the absence of nuclear staining?
Other comments and questions:
- Different cell lines were used for different experiments. For the sake of consistency, the brain metastatic cells and their parental cell line should be used throughout the paper.
- TEER: number of independent experiments and initial absolute values are missing.
- Do YUMM1.1-BrM4 cells form exclusively/primarily metastases in the brain or are other organs also targeted?
- The discussion is a repetition of the experiments performed rather than a critical comparison of the results to previous work.
Altogether, due to the limited novelty of the results, this paper does not meet the standards of Cancers journal.
Reviewer 2 Report
In this manuscript the authors demonstrated that the BBB integrity is important in limiting intercalation and extravasation of melanoma cells and therefore the formation of melanoma-brain metastasis.
The manuscript is of interest for the journal and is well written and clearly presented but it should be improved by minor revision.
Specific comments:
- Image resolution is low. Please provide higher resolution images
- In in vitro live cell imaging of melanoma intercalation into pMBMECs (figure 1G, supplementary video S1, Figure 2B-C and supplementary video S2, Figure 7 A and Supplementary video S3) it is possible to see only black areas presumably due to the displacement of the endothelial cells signal by melanoma cells. However, in these experiments also melanoma cells should be uniquely labeled and identified.
Reviewer 3 Report
The authors use cell culture models of transendothelial invasion and a zebrafish model of extravasation to examine the roles of IL-1b induction in migration of melanoma cells across the blood-brain barrier. They use syngeneic mouse cells for melanoma and endothelial cells. The authors find that induction of VCAM-1 and reduced junctional barrier function both contribute to increased transendothelial migration of melanoma cells in their models. This project represents a great deal of work in multiple model systems and reaches important conclusions for the field of transmigration, although the limitations of the models impact the novelty of the findings.
1. Mouse melanoma cells differ significantly from human melanoma samples genetically and phenotypically. Human brain endothelial cells and cell lines are commercially available, which asks the question of why all of these studies were performed with mouse cells. It is obviously not worth repeating the entirety of the project with human cells, but the authors should make clear that their model system is not an ideal reflection of human metastasis.
2. The authors use flow to discriminate between VCAM-dependent and independent transmigration. This is not a valid method for distinguishing the role of VCAM in transmigration. Blocking antibodies are available that prevent VCAM ligation with alpha4 integrin and have been used previously to block transendothelial migration. This would be a better way to study the dependence of transmigration on VCAM function.
3. The inclusion of sheer flow for these experiments significantly enhances the value of the authors findings, but then the authors remove the flow to study intercalation. Confocal characterization of intercalation under flow conditions would have been a truly novel observation, whereas intercalation without flow has been well-characterized in human melanoma cells without flow.
The authors’ in vivo mouse experiments are technically stunning, but PECAM-knock out humans are unknown with only SNP variants of PECAM in the human population. It is thus difficult to understand the relevance of the mouse experiments to human disease.
Reviewer 4 Report
This research article is titled "Compromised blood-brain barrier junctions enhance melanoma cell intercalation and extravasation." It discusses the study conducted on melanoma cells and their ability to penetrate the blood-brain barrier and spread to the brain. The authors have explored the mechanisms involved in melanoma cell intercalation and extravasation and the impact of compromised blood-brain barrier junctions on these processes. The research provides detailed information about the experimental methods used, including microscopy techniques and data analysis. The findings of the study have implications for understanding the metastatic behavior of melanoma and may contribute to the development of targeted therapies.
Key points
- The study focuses on the intercalation and extravasation of melanoma cells in the context of the blood-brain barrier.
- It describes the experimental techniques used, such as confocal microscopy and image analysis software.
- The study investigates the impact of compromised blood-brain barrier junctions on melanoma cell spreading and penetration.
- The research findings have implications for understanding the metastatic behavior of melanoma and may contribute to the development of targeted therapies.
Few minor suggestions to improve the paper:
- Introduction: The study for sure provides valuable insights into compromised blood-brain barrier junctions and melanoma cell intercalation. To enhance the paper's impact, it would be beneficial to provide a brief overview of the significance of the blood-brain barrier and the role of intercalation in melanoma metastasis.
- Methods: While the methods section provides sufficient details on the experimental setup and imaging techniques, including a flowchart or diagram illustrating the workflow would be helpful. This visual representation would aid readers in understanding the experimental process more easily.
- Results: The results section presents melanoma cell intercalation and extravasation findings. To enhance clarity, it would be beneficial to include statistical analysis and quantification of the results in legend as it would strengthen the paper's scientific rigor and help the reader go back and forth not to draft.
- A few grammatical errors throughout the paper require attention.
- A few grammatical errors throughout the paper require attention.
Round 2
Reviewer 1 Report
The revised version of the manuscript entitled “Compromised blood-brain barrier junctions enhance melanoma cell intercalation and extravasation” by Saltarin et al. has significantly improved in the interpretation of the data obtained in comparison to the literature. The limited novelty of the paper is still a major concern; however, the effort of the authors to address all my comments could be appreciated.
However, according to the supplementary tables, the TEER values of the primary mouse brain microvascular endothelial monolayers reached the plateau values of approx. 20 ohms x cm2 (before subtraction of the values of the empty filters?), which is surprisingly and unacceptably low. This value should be 100-150 for confluent primary mouse endothelial in mono-culture (doi. 10.1016/j.ejps.2010.11.005, doi: 10.1371/journal.pone.0236770, and several other papers). Since TEER reflects the integrity of the tight junctions, this in vitro model is by no means suitable to study phenomena that affect the junctions. The proper model would be an absolute prerequisite to accepting the validity of the in vitro data presented; therefore, the paper must be rejected in its present form.
Other comments:
- Fig. 3: N = 1 experiment is not acceptable.
- For the sake of consistency and novelty, the brain metastatic cells and their parental cell line should be used throughout the paper.
In conclusion, this paper does not meet the standards of Cancers.
Author Response
I am sorry to learn that the Reviewer understands level of TEER values as a measure of the quality of a BBB model. In the global BBB community, pMBMECs with their moderate TEER values are considered an established and valuable model of mouse BBB. However, I am also very grateful to have this pointed out to me, as it may be important to better explain the characteristics and conditions of pMBMECs. Various parameters have an enormous influence on the TEER values:
- The filter inserts.
Shayan et al. 2011 (doi.org/10.1016/j.ejps.2010.11.005) used Transwell permeable supports from Corning with a pore density of 4 × 10^6 pores/cm^2. We used Greiner Bio-One ThinCert TM, with a pore density of 1 x 10^8 pores/cm^2. - The measurement device
We used the CellZScope instrument known for its rather moderate TEER values. - The temperature.
Slight temperature differences or even measurements at room temperature change the TEER values enormously. All values given by us are measured at 37 °C. - Media
Thomson et al. 2021 (doi.org/10.1371/journal.pone.0236770) complemented the growth medium with 250 μM 8-(4-Chlorophenylthio)adenosine 3′,5′-cyclic monophosphate sodium salt (CTP-cAMP), 17.5μM RO-1724 (RO) (A cell-permeable, selective inhibitor of cAMP-specific phosphodiesterase), and 550nM hydrocortisone (HC). We consider these substances as unphysiological and therefore omit using them.
Moreover, TEER values that are too high could indicate an epithelial character of a barrier. BBB models with epithelial features are not suitable for studying extravasation of immune cells or metastatic cancer cells because they do not express the full range of endothelial cell adhesion molecules (e.g. VCAM-1) involved in this process.
We thank Reviewer 1 for his/her critical comments and have made the following changes to the 2nd revised manuscript. All changes in this 2nd revision are shown in green font.
To address the reviewers' criticism regarding the moderate TEER values the following changes have been made.
In lines 318 - 324, the following information was added to the method of TEER measurement.
- (Filter insert specification…) 1 x 108 pores/cm2 pore density
- (pMBMECs were grown…) without TEER-increasing media supplements such as hydrocortisone or cAMP-stabilizing agents.
- During all measurements, the instrument was placed in the cell-culture incubator to ensure the physiological temperature of 37 °C.
- The raw TEER values are listed in the Supplementary Tables 1 and 2.
Lines 778 - 786 now provide additional information about the pMBMECs to the discussion and point out the moderate TEER values (line 785):
“We have previously studied the BBB characteristics of pMBMECs in detail. pMBMECs closely mirror the RNA transcriptome of naïve brain microvessels including the expres-sion of BBB solute carriers and efflux transporters [30]. They are responsive to inflamma-tory stimuli with upregulation of cell adhesion molecules and temporary changes in bar-rier properties. The adherens junction protein VE-cadherin, the tight junction proteins claudin-5 and occluding, the junctional protein PECAM-1 and the junction associated proteins ZO-1 and ZO-2 were localized to the cell junctions of pMBMECs[22, 29, 44]. pMBMECs form moderate but significant TEER without prior stimulation of intracellular cAMP or culture in the presence of hydrocortisone [30, 31, 36].”
To address the reviewers' criticism that Fig. 3 is not acceptable the following additional changes have been made.
The chapter “Rescue of melanoma cell-induced BBB disruption by GM6001” has been dissolved. In the revised manuscript the respective experiment and findings are described in the chapter before, entitled “Melanoma cells intercalate into pMBMECs exclusively at their cell-cell junctions” (lines 513 – 528). The former Figure 3 has been moved to Supplementary Data and is now called Supplementary Figure 1. Reference is made to the "pilot" nature of this experiment in line 517.
Accordingly, the corresponding sentence in the simple summary has been moved to the sentence describing the junctional pathway (lines 39 - 41). To reduce the emphasis on novelty, the role of proteases is now "confirmed" - instead of "determined" as before (line 40 and line 51).
Supplementary Tables 1 and 2 show additional calculations to subtract the values of the empty filters. The data shown in Supplementary Figure 1 have been revised accordingly.
To account for these changes in numbering, Figures 4 through 9 have been renamed Figures 3 through 8 and Supplementary Figures 1 through 3 have been renamed Supplementary Figures 2 through 4 throughout the manuscript.